# P2X7 Is Involved in the Mouse Retinal Degeneration via the Coordinated Actions in Different Retinal Cell Types

**DOI:** 10.3390/antiox12010141

**Published:** 2023-01-06

**Authors:** Ponarulselvam Sekar, George Hsiao, Yuan-Shen Chen, Wan-Wan Lin, Chi-Ming Chan

**Affiliations:** 1Graduate Institute of Medical Sciences, Taipei Medical University, Taipei 110301, Taiwan; 2Department of Pharmacology, College of Medicine, National Taiwan University, Taipei 100233, Taiwan; 3Department of Pharmacology, College of Medicine, Taipei Medical University, Taipei 110301, Taiwan; 4Department of Neurosurgery, National Taiwan University, Yunlin Branch, Yunlin County 640203, Taiwan; 5Department of Ophthalmology, Cardinal Tien Hospital, New Taipei City 23148, Taiwan; 6School of Medicine, Fu Jen Catholic University, New Taipei City 242062, Taiwan

**Keywords:** P2X7, sodium iodate, retinopathy, microglia, photoreceptors

## Abstract

Adenosine triphosphate (ATP) released from dying cells with high concentrations is sensed as a danger signal by the P2X7 receptor. Sodium iodate (NaIO_3_) is an oxidative toxic agent, and its retinal toxicity has been used as the model of dry age-related macular degeneration (AMD). In this study, we used NaIO_3_-treated mice and cultured retinal cells, including BV-2 microglia, 661W photoreceptors, rMC1 Müller cells and ARPE-19 retinal epithelial cells, to understand the pathological action of P2X7 in retinal degeneration. We found that NaIO_3_ can significantly decrease the photoreceptor function by reducing a-wave and b-wave amplitudes in electroretinogram (ERG) analysis. Optical coherence tomography (OCT) analysis revealed the degeneration of retinal epithelium and ganglion cell layers. Interestingly, P2X7^−/−^ mice were protected from the NaIO_3_-induced retinopathy and inflammatory NLRP3, IL-1β and IL-6 gene expression in the retina. Hematoxylin and eosin staining indicated that the retinal epithelium was less deteriorated in P2X7^−/−^ mice compared to the WT group. Although P2X7 was barely detected in 661W, rMC1 and ARPE-19 cells, its gene and protein levels can be increased after NaIO_3_ treatment, leading to a synergistic cytotoxicity of BzATP [2′(3′)-O-(4-benzoylbenzoyl)adenosine-5′-triphosphate tri(triethyleneammonium)salt] and NaIO_3_ administration in ARPE-19 cells. In conclusion, the paracrine action of the ATP/P2X7 axis via cell–cell communication is involved in NaIO_3_-induced retinal injury. Our results show that P2X7 antagonist might be a potential therapy in inflammation-related retinal degeneration.

## 1. Introduction

P2X7 is a ligand-gated ion-channel receptor that is ubiquitously expressed in the mouse organism, including myeloid cells. The physiological ligand of P2X7 is adenosine triphosphate (ATP) that accumulates in a concentration range of mM at sites of tissue injury and inflammation. Thus, a high concentration of extracellular ATP acts as a very early endogenous damage-associated molecular pattern (DAMP) for P2X7 activation and inflammation induction [1]. Activation of P2X7 can rapidly trigger K^+^ and Ca^2+^ ion movement across the plasma membrane [2,3], leading to several cellular events, including NACHT, LRR and PYD domain-containing protein 3 (NLRP3) inflammasome activation, pannexin 1 and connexin hemichannels openings, membrane blebbing, reactive oxygen species (ROS) production, mitochondrial membrane potential loss, and eventually cell death [4,5,6,7].

The retina is a sensory tissue that is organized by diverse cell types in different cell layers with microcircuits [8]. Different retinal cells in different layers work together to encode visual information. The retinal pigment epithelium (RPE), a monolayer of pigmented cells, is located between photoreceptor cells and Bruch’s membrane. It is a critical component of the blood-retinal barrier, and plays an essential role in maintaining retinal homeostasis and promoting the health of photoreceptors. The importance of the RPE relies on its intimate relationship with light-sensitive photoreceptors, which allows transmission of information, transfer of nutrients to photoreceptors and phagocytosis of metabolic waste [9]. Thus, structural or functional injuries of RPE can trigger pathologic events in retinal diseases, such as age-related macular degeneration (AMD).

Inflammation contributes to the pathogenesis of many retinal degenerative diseases. Microglia, the primary resident immune cell type, constitute a key population of glia in the retina. Accumulating evidence indicates that microglia not only interact closely with synapses to maintain synaptic electrophysiological response to light but also constitute a crucial defense system in the retina to support their surrounding neural tissue. Emerging studies demonstrate that robust microglial activation is a disturbance in the cell–cell communication between microglia and other retinal cells and can manifest detrimental actions [10,11]. This is evidenced, at least from one example, by observing activated microglial cells prior to the initiation of photoreceptor death in animal models of retinitis pigmentosa [12]. In addition, activated microglia can secrete inflammatory cytokines, the key contributing factors for progressive cell death, to further aggravate retinal injury and loss of vision. For example, interleukin-1β (IL-1β) is pivotal in the mediation of innate immunity and contributes to several retinal diseases, including AMD, diabetic retinopathy and retinitis pigmentosa, as well as glaucoma [13].

In the retina, P2X7 was found in different layers of cells [14]. Recent studies indicate that P2X7 might be a therapeutic target for AMD, diabetic retinopathy and glaucoma [15,16,17]. Increased extracellular ATP levels were found in the vitreous samples of AMD patients with subretinal hemorrhage compared to the normal controls [18]. In mice, P2X7 is increased in the retina as early as one month following diabetes induction [19], as well as in the ischemic retina [20]. P2X7 blockade can attenuate progression of both diabetic neuronal and vascular pathology, such as retinal vascular permeability increase, vascular endothelial growth factor (VEGF) accumulation and interleukin-6 (IL-6) expression [19,21]. P2X7 is also involved in glaucoma, possibly through the induction of retinal ganglion cell death [22]. In the retina, although P2X7 expression was found in microglia [23], photoreceptors [18], RPE [15] and Müller glial cells [24], there is no comprehensive information to fully understand the cell-type-specific actions of P2X7 in the retina. Sodium iodate (NaIO_3_) is an oxidative toxic agent, and it has been widely used to study the pathogenesis of dry AMD [25,26,27,28,29,30]. In this study, we used NaIO_3_-treated mice and different retinal cells to understand the roles of the involvement of P2X7 in retinal degeneration.

## 2. Materials and Methods

### 2.1. Animals

Wild-type (WT) (C57BL/6J) mice were purchased from the Laboratory Animal Centre, National Taiwan University. P2X7^−/−^ mice on the C57BL/6J background were obtained from Jackson Laboratory (Bar Harbor, ME, USA). All animals were bred under specific pathogen-free conditions and maintained under a 12 h on-off lighting cycle at the National Taiwan University College of Medicine Laboratory Animal Centre. The animal experiments were conducted in accordance with institute regulations after receiving approval from the Institutional Animal Care and Use Committee, National Taiwan University College of Medicine (No. 20210290). For in vivo experiments, 6-week-old male mice were anesthetized by intraperitoneal shots of 50 mg/kg ketamine in combination with the relaxing agent of 10 mg/kg xylazine. NaIO_3_ was administered as a single intraperitoneal (i.p.) injection (25 mg/kg) as previously described [27,30]. After the injection, the mice were returned to the colony and kept under standard conditions. Retinal functions and structures were accessed after NaIO_3_ treatment for 3 days.

### 2.2. Scotopic Electroretinogram (ERG) Analysis in Mice

At day 3 post-injection, the mice were subjected to evaluate the rod and cone photoreceptor responses using ERG analysis. The ERG system is composed of an MP-36 4-channel amplifier and acquisition system (Biopac Systems, Inc., Pershore, UK) connected to a PS33-PLUS photic stimulator (Grass Technologies, Warwick, RI, USA). ERG measurements were recorded using 10 ms flash stimuli with an intensity of 16 (19.1 cd.s/m^2^). The amplitudes of a-wave and b-wave and implicit time were calculated from the ERG response [31].

### 2.3. Spectral-Domain Optical Coherence Tomography (SD-OCT) Imaging in Mice

SD-OCT was performed to assess the retinal morphology over the posterior pole in live experimental animals. The Micron III intraocular imaging system (Phoenix Research Labs, Pleasanton, CA, USA) is composed of an OCT engine and a scanning lens.

### 2.4. Histology Analysis

The mice were sacrificed by cervical dislocation, and eyes were enucleated, fixed at room temperature using Davidson solution (glacial acetic acid, 95% alcohol, 10% formalin and double-distilled water in a ratio of 4:12:5:15). After 24 h, retinal tissues over the posterior pole were embedded in paraffin, sectioned of 5 μm thickness using a microtome, processed in a standard manner, and stained with hematoxylin and eosin (H&E), periodic acid-Schiff (PAS) and Masson trichrome staining. The histological slides were examined for pathological changes by using the Olympus BX61 fluorescence microscope (Olympus Corp., Tokyo, Japan) with the MicroPublisher, QIMAGING MP3.3-RTV-R-CLR-10-C image system. The magnification powers were 100× and 400×.

### 2.5. Measurement of Cytokine Concentrations by Enzyme-Linked Immunosorbent Assay (ELISA)

Three days after NaIO_3_ injection, we carefully collected the blood from the mice through the heart by exsanguination. Preparation of serum was done to check the amounts of secreted cytokines by specific ELISA kits (R&D Systems, MN, USA), including IL-1β (R&D DY401-05), tumor necrosis factor-α (TNFα) (R&D DY410-05) and IL-6 (R&D DY406).

### 2.6. Cell Culture

Adult human RPE cell line ARPE-19 purchased from Food Industry Research and Development Institute (Hsinchu, Taiwan) was maintained in Dulbecco’s Modified Eagle Medium/Nutrient Mixture F-12 (DMEM/F12) supplemented with 10% fetal bovine serum (GibcoBRL, Invitrogen Life Technologies, Carlsbad, CA, USA), 100 units/mL penicillin and 100 μg/mL streptomycin (Sigma-Aldrich Co., St. Louis, MO, USA). Murine immortal microglial cell line BV-2, mouse photoreceptor cell line 661W and rat retinal Müller cell line rMC1 cells were cultured in complete high-glucose DMEM containing 4 mM L-glutamine and 25 mM glucose supplemented with 10% FBS, 3.7 g/L NaHCO_3_, 100 U/mL penicillin and 100 μg/mL streptomycin. 661W cells and rMC1 cells were given by Dr. Chih-Wen Shu (Institute of Biopharmaceutical Sciences, National Sun Yat-sen University, Kaohsiung, Taiwan) and Dr. Chang-Hao Yang (Department of Ophthalmology, National Taiwan University Hospital, Taipei, Taiwan). For most of the experiments, cells reaching 90–95% confluence were starved and synchronized in serum-free DMEM overnight before they were subjected to further experiments.

### 2.7. Flow Cytometry Analysis

After indicated treatment, cells were collected and washed with ice-cold phosphate-buffered saline (PBS). Cells were stained with Annexin V-FITC/PI and analyzed by flow cytometry (FACSCalibur, BD, Franklin Lakes, NJ, USA) according to the manufacturer’s instructions (BioLegend, San Diego, CA, USA). The cells in the respective quadrants were calculated by using CellQuest PRO software.

### 2.8. Quantitative Real-Time Polymerase Chain Reaction (PCR)

The expression of genes encoding IL-1β, IL-6, NLRP3, caspase-3, caspase-7, caspase-8, P2X7 and β-actin were determined by real-time PCR analysis with specific primers (Appendix A)Three days after NaIO_3_ i.p. injection, WT and P2X7^−/−^ mice were sacrificed to collect the whole retina. The whole retina was homogenized with 200 μL of TriPure isolation reagents (Roche Applied Science), total RNA was extracted and 1 μg of total RNA was reverse transcribed with an RT-PCR kit (Promega) according to the manufacturer’s instructions. Real-time PCR was performed in 96-well plates with the Fast Start SYBR Green Master. Each 25 μL PCR well contained complementary DNA (cDNA), Master Mix, gene-specific primers, and passive reference dye (ROX) to normalize the signals from the SYBR Green double-stranded DNA complexes during the analysis and to correct for well-to-well variations. PCR products were measured with an ABI Quant Studio 5 (Applied Biosystems, Foster City, CA, USA).

### 2.9. Immunoblotting

After treatments with the indicated drugs, cells were harvested, and equal amounts of the soluble protein were loaded and electrophoresed on sodium dodecyl-sulfate polyacrylamide gel electrophoresis (SDS-PAGE) and then transferred to Immobilon-P (Millipore, Billerica, MA, USA). Different proteins were determined by immunoblot analysis using antibodies for P2X7 extracellular (ecto) (APR-008 Alomone, Jerusalem, Israel), P2X7 (c-term) (APR-004 Alomone, Jerusalem, Israel), β-actin (sc47778 Santa Cruz, CA, USA) and enhanced Chemiluminescence Reagent Plus (Perkin Elmer, Wellesley, MA, USA).

### 2.10. Determination of Nicotinamide Adenine Dinucleotide Phosphate (NADPH) Level

NADPH was determined in retinal tissues using the kit from Abcam (ab65349) according to the manufacturer’s instructions. Briefly, 50 mg of tissue weight was lysed in extraction buffer provided using a Dounce homogenizer. A total of 200 μL of lysates was heated at 60 °C for 30 min to decompose NADP^+^ while NADPH was intact. The enzymatic activity was assayed by continuously monitoring the increase in NADPH absorbance at 340 nm for 30 min at 25 °C using colorimetry analysis.

### 2.11. Statistical Analysis

Data were expressed as mean ± standard error of the mean (SEM). Multiple groups were compared by one-way analysis of variance and Bonferroni post-test, making use of GraphPad software (San Diego, CA, USA). Two groups were compared with an unpaired Student’s t test and two-tailed *p*-value. Results were considered statistically significant when *p* < 0.05.

## 3. Results

### 3.1. NaIO_3_-Induced Retinopathy Is Alleviated in P2X7 Knockout Mice

ERG analysis was used to determine the changes in retinal functions in WT and P2X7^−/−^ mice. Before ERG recording, mice were kept in dark adaptation overnight. After NaIO_3_ injection for one day, the retinal functions recorded by ERG were still normal; however, the retinal functions of mice were totally lost after 5 days. Therefore, in our study, we conducted the ERG experiment in mice after receiving NaIO_3_ for 3 days. We found the a-wave amplitude, which is responsible for photoreceptor functions, and the b-wave amplitude, which is responsible for bipolar, amacrine and Müller cells, showed no significant difference between WT and P2X7^−/−^ control mice, even though the average b-wave amplitude was higher in P2X7^−/−^ mice (Figure 1A,B). After NaIO_3_ treatment, both the a- and b-wave amplitudes were obviously decreased in WT mice, but the NaIO_3_-induced retinopathy was alleviated in P2X7^−/−^ mice. Moreover, the a-wave and b-wave implicit time were concomitantly prolonged in NaIO_3_-treated WT mice, and both changes were reversed in P2X7^−/−^ mice (Figure 1C). Next, we performed an OCT experiment to understand the NaIO_3_-induced retinal degeneration on the posterior pole in P2X7^−/−^ mice. In NaIO_3_-treated WT mice, retinal thickness was decreased with derangements of the retinal structure, and layering from the outer nuclear layer (ONL) to the RPE layer was not clearly laminated (yellow arrow) (Figure 2A). Therefore, damages to photoreceptor cells and RPE cells are concluded. On the other hand, NaIO_3_-treated P2X7^−/−^ mice displayed a smaller decrease in retinal thickness, even though derangement of layering on the ONL and inner segment (IS)/outer segment (OS) was still shown.

In Figure 2B, the retinal structure around the optic disc is shown. Several hyperreflectivity foci over the ONL and RPE layers (orange arrows), as well as in the vitreous site (blue arrows), were found. Moreover, disruption on the RPE layer (yellow arrows) and decreased thickness between the ONL and RPE layers (red double arrowhead) were noticed. All these morphological changes observed in WT mice after NaIO_3_ treatment were not apparent in P2X7^−/−^ mice. The gross morphology of the retina in both WT and P2X7^−/−^ mice was also determined by H&E staining to ascertain the above findings seen in OCT. In WT mice, all layers of the retina were clearly visible and easily distinguished. Moreover, P2X7^−/−^ mice retina demonstrated a similar morphology. After NaIO_3_ treatment in WT mice, the retinal structure was disturbed. Some round, pigmented granules lying along with the RPE layer were observed (Figure 3). The granules were irregular in size and shape, and the pigmentations were homogenously distributed within the granules, which showed a similar dark color to the pigments within the RPE layer (black arrows). Since hemosiderin is usually more heterogenous and the vitreous as well as retinal hemorrhages were not seen, hemosiderin was excluded. Thus, melanin-containing pigment granules are considered. Moreover, retinal thinning and irregular lining with diffuse cell loss of bipolar, amacrine and horizontal cells in the inner nuclear layer (INL) and photoreceptor cells in the ONL were observed. The cell bodies of the INL and ONL became enlarged and irregularly arranged. In P2X7^−/−^ mice, no retinal granules lying along the RPE layers were observed. Moreover, decreases in ONL and INL thicknesses and reversal of the irregular lining of the ONL and INL were noticed.

To exclude the possibility of other materials accumulated within the pigment granules, PAS staining and Masson trichrome staining were applied in the retinal sections (Figure 4). PAS staining was used to detect glycogen, glycoproteins, glycolipids and mucopolysaccharides (mucins and hyaluronic acid) in tissues, whereas Masson trichrome staining was used to identify collagen deposition. Our results showed that the pigment granules were all negative for PAS (Figure 4A) and Masson trichrome staining (Figure 4B). There were no positive pink spheroids in PAS staining nor positive blue granules in Masson trichrome staining after NaIO_3_ injection, except the appearance of dark granules as we mentioned before in Figure 3 (black arrows). These data suggest that the pigment granules are not composed of glycogen, hyaluronic acid or collagen. In addition, the thicknesses of the ONL and INL in NaIO_3_-treated WT mice were decreased and the irregular alignments were apparent. The NaIO_3_-treated P2X7^−/−^ mice showed the relatively normal thickness of the ONL and INL, and no melanin-containing granule deposition.

### 3.2. P2X7 Knockout Reduces NLRP3, IL-1β and IL-6 Gene Expression in Retinal Tissues after NaIO_3_ Injection

The effects of NaIO_3_ on gene expression relating to inflammation were measured in the isolated retina. Treatment with 25 mg/kg NaIO_3_ for 36 h triggers inflammatory responses in the retina of WT mice, as evidenced by the increased mRNA levels of NLRP3, IL-1β and IL-6. In contrast, P2X7^−/−^ mice displayed a significant attenuation of these gene expressions after NaIO_3_ treatment (Figure 5A). Furthermore, we collected the serum after 3 days of NaIO_3_ injection, and measured IL-1β, TNFα and IL-6 protein levels using ELISA. We found that only the IL-6 protein level was increased in the WT NaIO_3_ group, and this response was reduced in the P2X7^−/−^ NaIO_3_ group (Figure 5B).

### 3.3. P2X7 Knockout Reduces Executional Caspases Expression in Retinal Tissues after NaIO_3_ Injection without Affecting NADP/NADPH

NaIO_3_-induced oxidative stress and caspase activation are involved in RPE/photoreceptor damage [28,30,32]. So, we were interested in understanding the caspase expression and oxidative stress upon NaIO_3_-induced retinal degeneration. We found that NaIO_3_ can significantly trigger caspase-3, -7 and -8 gene expressions in WT mice. In contrast, such caspases upregulation was significantly reduced in P2X7^−/−^ mice (Figure 6A). For oxidative stress, we found that NaIO_3_ did not change the cellular levels of total NADP, NADPH and ratio of NADP/NADPH in the retinal tissues of WT mice. The contents of these molecules were neither changed in P2X7^−/−^ mice (Figure 6B).

### 3.4. P2X7 Is Differentially Expressed in Retinal Cells and P2X7 Activation Increases NaIO_3_-Induced Cytotoxicity

After showing the retinal protection in P2X7^−/−^ mice, the mechanisms of the involvement of P2X7 in the cytotoxicity induced by NaIO_3_ were elaborated. We first determined the expression level of P2X7 and compared the death effects of BzATP (a selective P2X7 agonist) in 661W photoreceptor, ARPE-19 retinal pigment epithelial, BV-2 microglial and rMC1 Müller cells. We found that P2X7 protein was highly expressed in BV-2 cells, but marginally detected in 661W, rMC1 and ARPE-19 cells (Figure 7A). Moreover, we showed that P2X7 mRNA (Figure 7B) and protein (Figure 7C) levels can be significantly increased in ARPE-19, rMC1 and 661W cells after NaIO_3_ treatment. In agreement with the protein expression of P2X7, after treatment of BzATP (200 μM) for 18 h, a severe cell death was observed in BV-2 cells, but not in 661W, rMC1 and ARPE-19 cells (Figure 7D).

To further interpret the P2X7-dependent cell–cell communication under NaIO_3_ stress conditions, we determined the combinational death effects of P2X7 agonist BzATP and NaIO_3_ treatment. We found that NaIO_3_ at 30 mM can induce cell death in all cell types, while BzATP (200 μM) alone only can induce cytotoxicity in BV-2 cells (Figure 8A). As the P2X7 activation itself cannot induce significant cell death in 661W, rMC1 and ARPE-19 cells, we supposed the effects resulting from P2X7 activation in individual cell types might be unmasked under the pathological stress condition, such as the retinal degeneration in NaIO_3_-treated mice. Therefore, we determined the effects of the agonist and antagonist of P2X7 together with NaIO_3_. We found that BzATP can enhance NaIO_3_-induced cytotoxicity in ARPE-19 cells but not in BV-2, 661W and rMC1 cells (Figure 8A). On the other hand, when treating A438079 (a selective P2X7 antagonist), the effects of NaIO_3_ in 661W, rMC1, ARPE-19 and BV-2 cells were not changed (Figure 8B). These data suggest that there is no autocrine effect of P2X7 under NaIO_3_ treatment in these cell types. However, BzATP and NaIO_3_ exert an enhanced cytotoxicity in ARPE-19 cells (Figure 8A). These findings further suggest that ATP released from other damaged cell types in the retina might be able to amplify the death outcome in stressed ARPE cells.

## 4. Discussion

Many inflammatory diseases result from excessive activation of the immune system by initiating IL-1β, activating NF-κB and inducing cell death signaling. In the retina, the P2X7 receptor is present in both inner and outer cells [14]. Using a receptor antagonist and KO mice, P2X7 is established as a therapeutic target for treating retinal diseases [16], including AMD [15,18], diabetic retinopathy [19,21] and glaucoma [22,33]. Antagonists of P2X7 prevent ATP-induced neuronal apoptosis in glaucoma, diabetic retinopathy and AMD [19,21,34]. Despite the therapeutic potential of the P2X7 blockade in retinal diseases and the extensive distribution of P2X7 in various retinal cell types has been documented, there are no comprehensive results to understand the cell-type-specific action of P2X7 in the retina. In this study, we clearly demonstrate the pathological role of P2X7 in NaIO_3_-induced retinopathy by using WT and P2X7^−/−^ mice. In this study, we identified microglia express the highest P2X7 than photoreceptors, Müller cells and RPE cells. Therefore, through cell–cell communication, microglia may be the first cells to sense retinal injury-induced ATP expression and elicit P2X7-mediated inflammation. Afterwards, this inflammation can further amplify the cell death circuit in the retina.

Our study showed the P2X7 blockade is beneficial in NaIO_3_-induced retinopathy functionally and morphologically. The functional data from ERG and OCT analyses, as well as histological examinations from H&E staining, all indicate that the retina damages induced by intraperitoneal injection of NaIO_3_ are alleviated in P2X7^−/−^ mice. In NaIO_3_-induced retinal degeneration, the disturbed ONL layering in SD-OCT analysis, the decreased a-wave amplitude in ERG examination, and the ONL thinning in H&E staining clearly signify photoreceptor damage. In addition, the decreased b-wave amplitude in ERG analysis indicates synaptic damage and synaptic transmission impairment between the photoreceptor and bipolar cells after intraperitoneal injection of NaIO_3_. The slightly higher b-wave amplitude in P2X7^−/−^ mice is consistent with a previous study and suggests that P2X7 can regulate rod and cone cell signaling [35]. After NaIO_3_ stimulation, we also found the disrupted RPE layer and melanin aggregation along the RPE layer are attenuated by P2X7^−/−^.

The appearance of pigment granules under NaIO_3_ treatment is associated with retinal degeneration [26]. In the retinal tissue, melanin pigments and hemosiderin should be considered. However, hemosiderin is shown to be more heterogenous and should be seen under the condition of the vitreous or retinal hemorrhage, while it is not present in this study. Our results showed that the pigmentations were homogenously distributed within the granules. Therefore, the melanin-containing granules are considered. This clumping of melanin may be due to RPE cell death. Since glycogen and collagen accumulation are commonly observed in the subretinal space and contribute to diabetic retinopathy and retinal fibrosis, respectively [36], we conducted PAS and Masson trichrome staining. PAS staining was used to detect polysaccharides such as glycogen, muco-substances such as glycoproteins, glycolipids, mucins and vitreal hyaluronic acid in tissues. Masson trichrome staining was used to distinguish collagen deposition. Our data exclude the accumulation of both substances in the retinopathy model of NaIO_3_.

Based on the above functional and morphological defects, we show the involvement of inflammation and cell death in the pathological processes of the retina. After NaIO_3_ injection for 36 h, gene expressions of caspase-3, -7, and -8, NLRP3, IL-1β, and IL-6 are significantly upregulated in the retina of WT mice but are significantly attenuated by P2X7^−/−^. The effects of P2X7 activation on neuroinflammation [37,38] and cell necrosis in microglia [7] have been demonstrated previously. Under different activation conditions, P2X7 mediates microglial activation, proliferation (microgliosis) and cell death. Therefore, P2X7 can lead to a deleterious cycle of neuroinflammation and neurodegeneration. In this study, we detect the upregulation of caspase-3, -7 and -8 gene expressions in the retina, and suggest the contribution of caspase-induced cell death in retinopathy. A previous study showed P2X7 activation can initiate multiple caspases, including caspase-1, -3 and -8, in cell death [39], while the P2X7 antagonist can reduce 6-OHDA-induced dopaminergic toxicity in rats through the suppression of the expressions of caspase-3 and -9 [40]. In our study, we found the changes in various caspases further strengthen the amplification circuit in neuroinflammation and cell death.

Until now, the expression and function of P2X7 in various retinal cell types remain controversial. P2X7 expression was found in microglia [23], photoreceptors [18], RPE [15] and Müller glial cells [24], and can be upregulated during vitreoretinopathy [41]. However, some studies did not detect P2X7 expression in Müller cells [42,43]. Previously, P2X7 activation has been shown to induce cell death in photoreceptors [18,44,45], cholinergic neurons [46], ganglion cells [47,48], RPE cells [15] and microglial cells [7] by forming a ligand-gated ion channel. However, some studies revealed that photoreceptor cell deaths are based on a non-selective P2 agonist ATP rather than a selective P2X7 agonist [18,49]. Several studies showed that P2X7-dependent cytotoxicity in retinal ganglion cells is induced under the conditions of H_2_O_2_ [50,51] and cannabinoids [52] stimulation, but not induced by P2X7 self-activation directly. In addition, BzATP has been shown to induce cell death in primary human RPE cells [53] and IL-1α-pre-treated ARPE-19 cells [54]. However, one study did not detect BzATP-induced cytotoxicity in ARPE-19 cells [55]. All the above findings inspire us to confirm again the P2X7-mediated death response in various retinal cell types by using BzATP. Moreover, even though NaIO_3_-induced retinal degeneration is regarded as an animal model of AMD [56], indicating its preference for RPE damage, there is no comparative study of its actions in various retinal cell types at this moment. Therefore, we compared the cellular responses of selective P2X7 agonist BzATP and NaIO_3_ administration in four retinal cell types (661W photoreceptors, ARPE-19, rMC1 Müller cells and BV-2 microglia) and determined the relatively major cell type(s) in the retina degeneration induced by P2X7 activation and NaIO_3_ treatment.

In this study, we found that P2X7 activation by selective agonist BzATP leads to cytotoxicity in a cell-type-specific manner, with a preference for microglia. This effect is correlated to their relative expression level of P2X7. Among the four cell types, BV-2 microglia express a much higher amount of P2X7 than 661W, rMC1 and ARPE-19 cells. On the other hand, NaIO_3_ can cause a similar cell death degree in all these cell types. Of note, BzATP can increase the cytotoxicity of NaIO_3_ only in ARPE-19 cells. We suggest this effect might be due to the upregulation of P2X7 expression in NaIO_3_-treated ARPE-19 cells, as well as other unclarified integration of intracellular death events in ARPE-19 cells. Even though the P2X7 gene and protein expression in 661W and rMC1 cells are also increased by NaIO_3_, we do not observe the additive death effect between BzATP and NaIO_3_ in both cell types. Therefore, we suggest that depending on the cell types and cellular context, P2X7 activation under inflammatory and stressed microenvironments, which is possibly orchestrated in a paracrine manner, can deteriorate tissue injury and disease pathology. A previous study showed that P2X7 upregulated by gp120 provides an inflammatory microenvironment that causes BV-2 cell death [57]. By using the P2X7-selective antagonist A438079, we exclude the autocrine action of ATP-P2X7 in the death event of these cell types, even though P2X7 might induce a positive autocrine feedback loop to control the inflammatory outcome in other tissues. This autocrine action of P2X7 is based on the observation that P2X7 expression is increased in fibrotic and injured liver tissues, and this effect is inhibited by A438079 [58]. As for the limitations of our study, we did not test whether macrophages and/or microglia are involved in the retinal tissue, as we showed that inflammation plays an important role in NaIO_3_-induced retinal degeneration. Moreover, we demonstrated the caspase activation in NaIO_3_-treated retina, but the detailed locations of apoptotic activity in the retina were not determined.

Our findings suggest P2X7 plays a key paracrine role in the amplification links between inflammation and cell death in the retina, which involves cell–cell communication of retinal cell types and leads to the progression of retinopathy. Such a scenario actually has been proposed by observing that P2X7 can be upregulated in retinal ganglion cells by activated Müller cells [59], and the P2X7 receptor antagonist can protect retinal ganglion cells by inhibiting microglial activation [59]. Likewise, ligation of CD40 in human Müller cells induces P2X7-dependent death of retinal endothelial cells [60]. Moreover, such a cell–cell communication event has been demonstrated in activated Müller cells-retinal ganglion cells [59], optic nerve ganglion cells [61], microglia retinal ganglion cells [62], etc. Besides cell death, microglia-derived inflammation contributing to retinopathy is an important factor. For example, NLRP3-derived IL-1β secretion from retinal microglia leads to photoreceptor neurodegeneration [63] and retinal ganglion cell death in chronic ocular hypertension [62].

## 5. Conclusions

Taken together, our findings in mice and various cell types demonstrate that both the cell-cell communication and paracrine action of ATP/P2X7 axis are important in retinal injury (Figure 9). Given that microglial death is much sensitive to BzATP than photoreceptors, RPE cells and Müller cells, we suggest that cell damage-released endogenous ATP is major in acting on the microglia to induce inflammation and even microglial death. Moreover, microglia-derived neuroinflammation orchestrates a deleterious microenvironment to cause retinal degeneration.

## Figures and Tables

**Figure 1 antioxidants-12-00141-f001:**
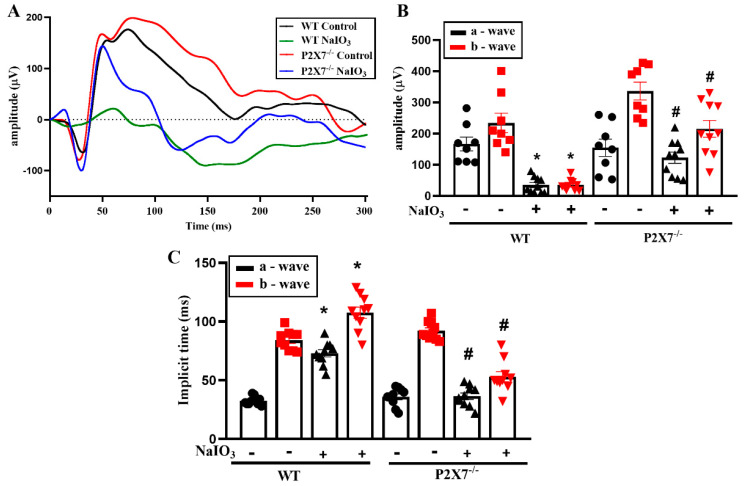
P2X7^−/−^ exerted protection against NaIO_3_-induced photoreceptor injury. (**A**) Representative scotopic ERG responses from WT and P2X7^−/−^ mice with or without NaIO_3_ (25 mg/kg) intraperitoneal injection. Three days after injection, ERG analysis was performed. (**B**) Quantification of the average amplitudes of a and b waves from WT control mice (*n* = 8), WT NaIO_3_-treated mice (*n* = 8), P2X7^−/−^ control mice (*n* = 8) and P2X7^−/−^ NaIO_3_-treated mice (*n* = 10). (**C**) Quantification of the implicit time of a and b waves from the 4 groups of mice. The data were presented as the mean ± SEM. * *p* < 0.05 compared with the WT control group, # *p* < 0.05 compared with the WT control group treated with NaIO_3_.

**Figure 2 antioxidants-12-00141-f002:**
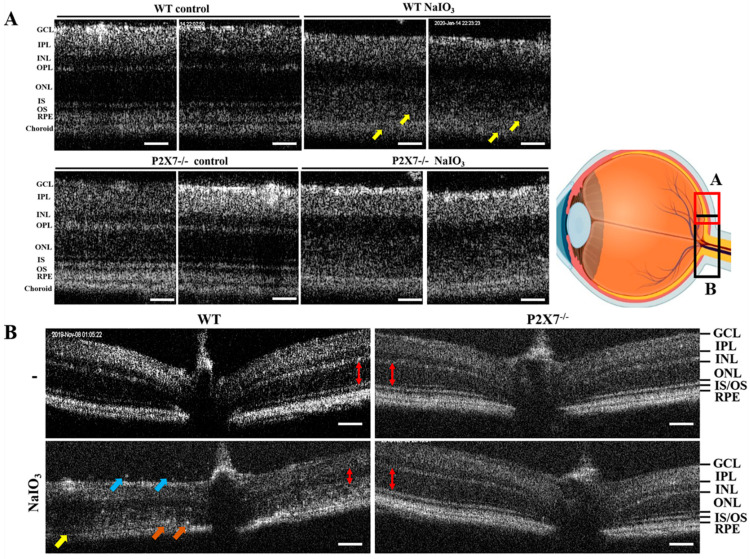
P2X7 knockout protected mice from NaIO_3_-induced retinal epithelium injury. OCT images in posterior pole region (**A**) and optical nerve area (**B**) revealed retinal pigment epithelium degeneration after the mice were treated with 25 mg/kg of NaIO_3_ for 3 days. RPE damage (yellow arrow in (**A**,**B**) was observed in the WT group but not in the P2X7^−/−^ group treated with NaIO_3_. Hyperreflective spots in both the vitreous region (blue arrow in (**B**)) and the retina (orange arrow in (**B**) were shown. The ONL thickness was marked by the red double arrowheads in (**B**). The mice numbers were 8, 8, 8 and 10 for WT control, WT NaIO_3_, P2X7^−/−^ control and P2X7^−/−^ NaIO_3_ groups, respectively. Scale bar: 200 μm. RPE, retinal pigment epithelium; OS, outer segment; IS, inner segment; ONL, outer nuclear layer; OPL, outer plexiform layer; INL, inner nuclear layer; IPL, inner plexiform layer; GCL, ganglion cell layer.

**Figure 3 antioxidants-12-00141-f003:**
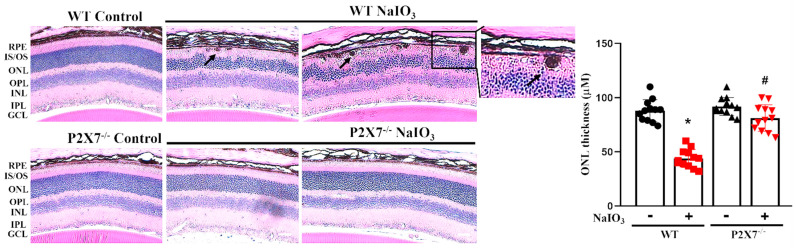
Histological H&E examination of retinal injury caused by NaIO_3_ was alleviated in P2X7^−/−^ mice. Retinal morphology and loss of photoreceptors and cell nuclei were evaluated by H&E by light microscopy at 3 days post-intraperitoneal injection of NaIO_3_ (25 mg/kg) in WT and P2X7^−/−^ mice. Shown were representative light microscopic images of H&E staining in retinal sections. Enlargement of cell bodies in the ONL and INL, pigment granules along the RPE layer (black arrows) and ONL thinning were observed. Quantitative analysis of ONL thickness was shown. The mice number was 12 for each group. Scale bar: 200 μm. * *p* < 0.05, indicating significant decrease in ONL thickness by NaIO_3._ # *p* < 0.05 compared with the WT control group treated with NaIO_3_.

**Figure 4 antioxidants-12-00141-f004:**
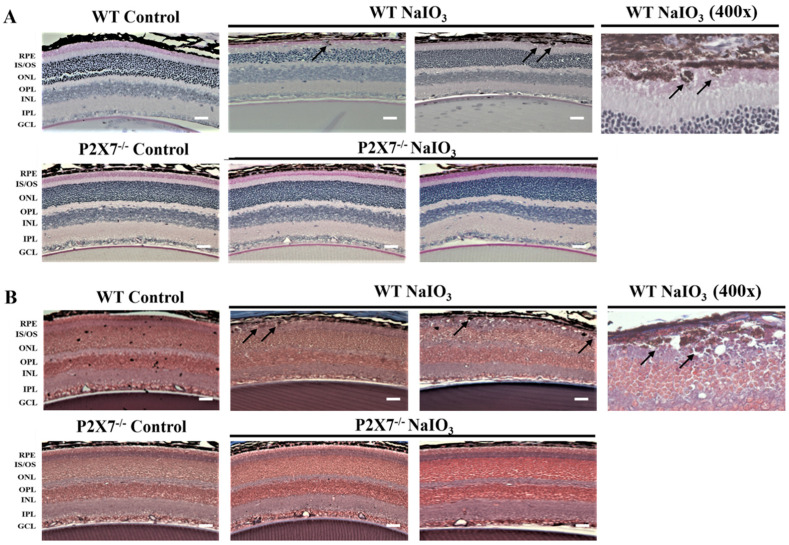
Histological examination of PAS and Masson trichrome staining. Representative light microscope images of PAS (**A**) and Masson trichrome (**B**) staining of retinal sections in WT and P2X7^−/−^ mice at 3 days post-intraperitoneal injection of NaIO_3_ (25 mg/kg) were shown. The visible retinal pigment granules were shown as black arrows in both PAS (**A**) and Masson trichrome (**B**) staining. The mice number for PAS staining was 5 for each group and for Masson trichrome staining was 7 for each group. Scale bar: 200 μm.

**Figure 5 antioxidants-12-00141-f005:**
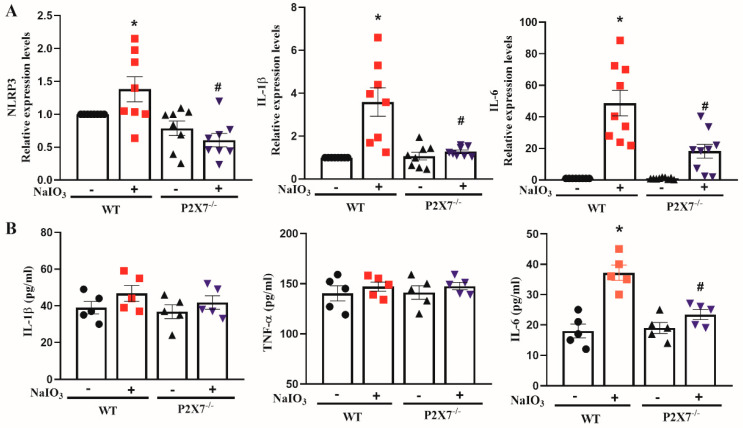
NaIO_3_-induced neuroinflammation in retina was reversed by P2X7 knockout. After 36 h post-injection of NaIO_3_ (25 mg/kg), mice retina was collected. (**A**) Total mRNA was extracted and reversely transcribed for quantitative PCR analysis of NLRP3, IL-1β and IL-6 mRNA expression. The mice number for qPCR analysis was 8 for each group. Values were normalized to β-actin gene expression and were expressed relative to the control group. (**B**) Parallelly, serum from mice was collected and subjected to ELISA to determine the amounts of IL-1β, TNF-α and IL-6, respectively. The mice number for ELISA was 5 for each group. The data were presented as the mean ± SEM. * *p* < 0.05 compared with the WT control group, # *p* < 0.05 compared with the WT control group treated with NaIO_3_.

**Figure 6 antioxidants-12-00141-f006:**
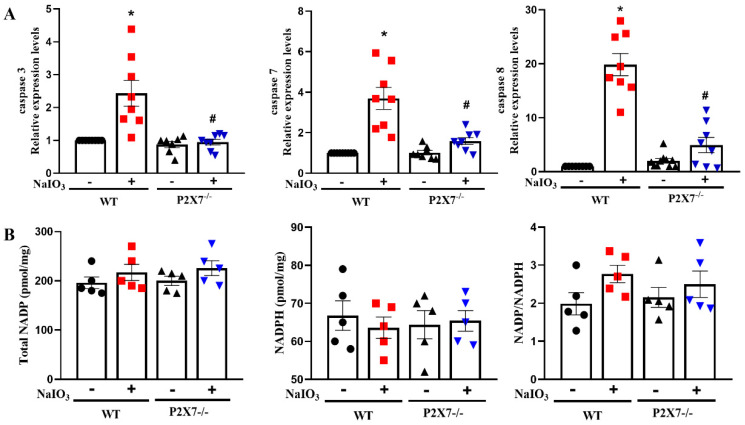
NaIO_3_-induced caspase expression in retina was reversed by P2X7 knockout^.^ (**A**) After 3 days post-injection of NaIO_3_ (25 mg/kg), mice retina was collected, and total mRNA was extracted and reversely transcribed for quantitative PCR analyses of caspase-3, caspase-7 and caspase-8 mRNA expression. The mice number for qPCR analysis was 5 for each group. Values were normalized to β-actin gene expression and were expressed relative to the control group. (**B**) Using the retinal tissues to measure the cellular contents of total NADP, NADPH and NADP/NADPH as per manufacturer’s instructions. The mice number for NADPH activity was 5 for each group. * *p* < 0.05 compared with the WT control group, # *p* < 0.05 compared with the WT control group treated with NaIO_3_.

**Figure 7 antioxidants-12-00141-f007:**
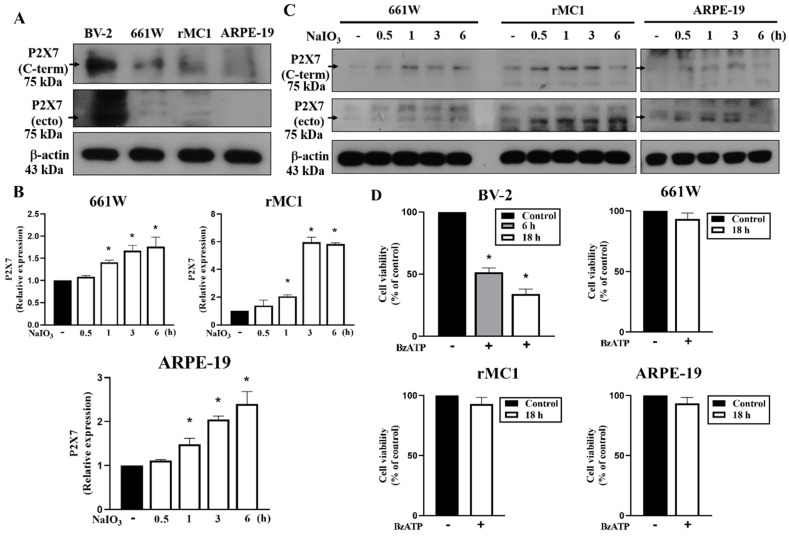
Comparison of P2X7 expression and effects of BzATP on cell viability in BV-2, 661W, ARPE-19 and rMC1 cells. (**A**) P2X7 protein level in each cell type was determined by immunoblotting. (**B**,**C**) Cells were treated with NaIO_3_ (30 mM) for the indicated times. P2X7 mRNA level was determined by real-time PCR (**B**) and P2X7 protein was determined by immunoblotting (**C**). (**D**) Each cell type was treated with vehicle or BzATP (200 μM) for 6 or 18 h. Cell death was determined by Annexin V/PI staining with FACS. * *p* < 0.05, indicating the increased P2X7 gene expression by NaIO_3_ (**B**) and the cell death caused by BzATP (**D**).

**Figure 8 antioxidants-12-00141-f008:**
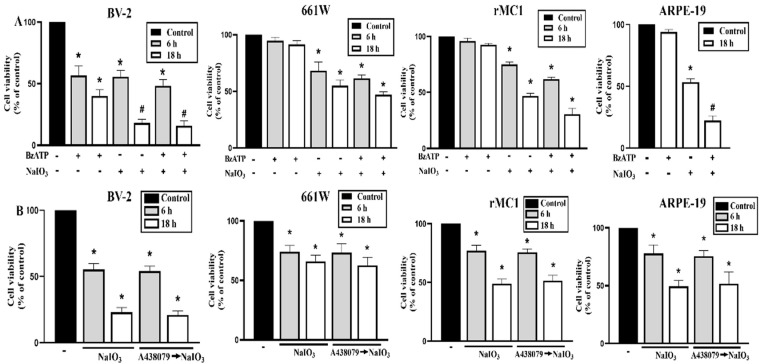
Effects of BzATP and A438079 on NaIO_3_-induced cell death in retinal cells. BV-2, 661W, rMC1 and ARPE-19 cells were treated with BzATP (200 μM) (**A**), A438079 (10 μM) (**B**) and/or NaIO_3_ (30 mM) (**A**,**B**) for 6 or 18 h. Cell death was determined by Annexin V/PI staining with FACS. * *p* < 0.05, significant inhibition of cell viability. # *p* < 0.05, an additive cytotoxicity was observed upon NaIO_3_ and BzATP co-treatment in ARPE-19 cells.

**Figure 9 antioxidants-12-00141-f009:**
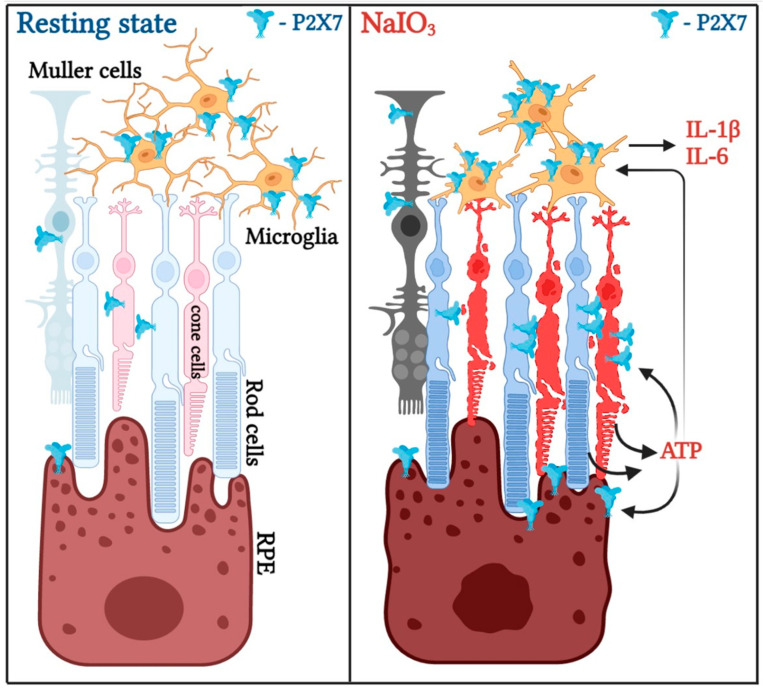
Schematic representation of P2X7 activation in amplification of NaIO_3_-induced neuroinflammation and retinal cell death. (**Left**) In normal conditions, P2X7 is constitutively and highly expressed in microglia than photoreceptors, RPE and Müller cells. (**Right**) Upon NaIO_3_ treatment, P2X7 expression in photoreceptors, Müller cells and RPE cells is increased, and cell death in these cell types, as well as microglia, is evoked. ATP releasing from the dying and damaged cells acts as a DAMP to activate microglia for inflammatory IL-1β and IL-6 gene expression. High concentration of ATP might also promote RPE cell death under NaIO_3_ stress.

## Data Availability

Data is contained within the article and Appendix A.

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
