# Peer review of "P2X7 Is Involved in the Mouse Retinal Degeneration via the Coordinated Actions in Different Retinal Cell Types"

_antioxidants, 2023, doi:10.3390/antiox12010141_

Round 1

Reviewer 1 Report (Previous Reviewer 1)

The revised manuscript has been improved according to reviewer's comments.

Author Response

Thank you for your reviewing the manuscript!

Reviewer 2 Report (Previous Reviewer 3)

Query 2 has been disregarded. Please modify the ms. accordingly. It reads:

It is somewhat contradictory that P2X7 is ubiquitously expressed (which means that is expressed in virtually all cell types) and in the same sentence particularize it for myeloid cells (lines 30-31). It is more appropriate to state that P27X is ubiquitously expressed in the mouse organism including myeloid cells.

The article is still in need of a thorough grammatical revision. Many incorrect or improvable expressions and sentences remain throughout the ms. The authors must have it revised in depth by an english-proficient colleage.

Author Response

Thank you for your reviewing. The first sentence has been corrected and the manuscript has been reviewed thoroughly.

This manuscript is a resubmission of an earlier submission. The following is a list of the peer review reports and author responses from that submission.

Round 1

Reviewer 1 Report

Chan et al
P2X7 enhances the retinal degeneration through the paracrine action of ATP/P2X7 axis in different retinal cells
Chan and colleagues found that
P2X7, receptor for ATP, enhance NaIO3-induced retinal cell death in vitro and in vivo. In contrast to wild type mice, P2X7-/- mice diminished the NaIO3-induced retinopathy and inflammatory NLRP3, IL-1 beta and IL-6 gene expression. P2X7 was elevated in different retinal cells in response to NaIO3 treatment and its agonist BzATP enhanced NaIO3-induced cell death.

Overall, the authors present novel and interesting findings. The manuscript is also well written.  There are some minor concerns need to be addressed as the followings:

1.     Most of NaIO3 injection for animal AMD model are through intravenous from either tail or orbital venous using higher dose (>40 mg/kg) of NaIO3 to see the retinal damage. The authors inject low dose of NaIO3 (25 mg/kg) via i.p. Does that mean i.p injection is more toxic or easier to have NaIO3 accumulation for retinal cells? The authors may need to explain why i/p injection is better than i.v injection.

2.     Does BzATP also enhance NaIO3-induced inflammatory cytokines expression in different ocular cells?

3.     P2X7 antagonist A438079 does not significantly rescue NaIO3-induced ocular cell death (Figure 8B, the “B” labeling is missing in figure). The authors may need to have discussion. Also, how do the authors conclude the effects are through paracrine action?

4.     The IRB statement section can be deleted in the manuscript, since no patient is involved in this study. 

Author Response

Chan et al P2X7 enhances the retinal degeneration through the paracrine action of ATP/P2X7 axis in different retinal cells. Chan and colleagues found that P2X7, receptor for ATP, enhance NaIO3-induced retinal cell death in vitro and in vivo. In contrast to wild type mice, P2X7-/- mice diminished the NaIO3-induced retinopathy and inflammatory NLRP3, IL-1 beta and IL-6 gene expression. P2X7 was elevated in different retinal cells in response to NaIO3 treatment and its agonist BzATP enhanced NaIO3-induced cell death.

Overall, the authors present novel and interesting findings. The manuscript is also well written.  There are some minor concerns need to be addressed as the followings:

  1. Most of NaIO3 injection for animal AMD model are through intravenous from either tail or orbital venous using higher dose (>40 mg/kg) of NaIO3 to see the retinal damage. The authors inject low dose of NaIO3 (25 mg/kg) via i.p. Does that mean i.p injection is more toxic or easier to have NaIO3 accumulation for retinal cells? The authors may need to explain why i.p. injection is better than i.v. injection.

Ans: Based on available papers, it seems that NaIO3 injection by i.p. or i.v. can induce similar effect on retinal degeneration. According to Yang et al. (Invest Ophthalmol Vis Sci 2014; 55:1696-705) and Young et al. (Mol Ther Methods Clin Dev. 2019; 14:113-125), 25 mg/kg NaIO3 by i.p. injection can induce significant morphological retinal degeneration. The effect occurs at 3-4 days after injection (Yang et al., 2017). Likewise, Ma et al. (Cell Death Dis, 2020; 11:24) and Tsai et al. (Biomedicines 2022;10:159) used 30 mg/kg of NaIO3 i.p. for 3 or 4 days and found deleterious effects of degeneration respectively. As to i.v. administration, Mogiruchi et al. (Invest Ophthalmol Vis Sci. 2018;59:3476-3487) injected mice with 20 mg/kg or 40 mg/kg of NaIO3 and they observed 40 mg/kg induces significant degeneration effects from 5th day.

  1. Does BzATP also enhance NaIO3-induced inflammatory cytokines expression in different ocular cells?

Ans: Our in vivo data showed the ability of NaIO3 to induce inflammatory cytokines in mice retina (Fig. 5). These are consistent to the findings of Ma et al. (2020). Currently we do not investigate the origins of these cytokines.  However, in our study NaIO3-induced cell death was enhanced by BzATP in BV-2 microglial and APRE-19 cells but not in 661W and rMC1 cells (Fig. 8). In addition, as we know microglia are the major cell types to induce large amounts of inflammatory cytokines and their death-derived damage molecular patterns further contribute to inflammation, so we think microglia are possibly the major cell types involving in P2X7-mediated inflammation in NaIO3 model. 

  1. P2X7 antagonist A438079 does not significantly rescue NaIO3-induced ocular cell death (Figure 8B, the “B” labeling is missing in figure). The authors may need to have discussion. Also, how do the authors conclude the effects are through paracrine action?

Ans: (1) We have added “B” labeling in Fig. 8B. (2) We more emphasized the paracrine role of ATP/P2X7 in Line 439 and 446. To properly indicate our findings, we have changed our manuscript title as “P2X7 involves in NaIO3-induced retinal degeneration via actions in different retinal cell types”.

  1. The IRB statement section can be deleted in the manuscript, since no patient is involved in this study. 

Ans: Yes, we have revised it according to the journal regulations.

Reviewer 2 Report

The authors examined a role of P2X7 receptor in retinal degeneration models in vivo and in vitro. The study is overall well-done and well-organized, and seems to provide novel information for mechanisms underlying dry-type AMD and diabetic retinopathy. But, I feel additional data would prove their hypothesis in this study. I have some concerns about this manuscript as follows:

1.    Please show the abbreviation of the term “BzATP” in abstract.

2.    Please demonstrate locations of apoptotic microglial cells in P2X7 -/- or wild-type mice treated with sodium iodate injections regarding figure 3/4 using double staining with TUNEL and microglial markers..

3.    As shown in figure 9, under the sodium iodate stimulation, please demonstrate how inflammatory cytokines in the supernatants derived/secreted from BV2 microglial cells involve cell death in cultured RPE cells.

Finally the authors should include recent review article looking at the role of P2X7 in retinal diseases (Biochem Pharmacol  doi: 10.1016/j.bcp.2022.114942.Epub 2022 Feb 5) to ensure the novelty of this study.

Author Response

Reviewer 2

The authors examined a role of P2X7 receptor in retinal degeneration models in vivo and in vitro. The study is overall well-done and well-organized, and seems to provide novel information for mechanisms underlying dry-type AMD and diabetic retinopathy. But, I feel additional data would prove their hypothesis in this study. I have some concerns about this manuscript as follows:

  1. Please show the abbreviation of the term “BzATP” in abstract.

Ans: As per your suggestion we included the abbreviation of BzATP on the abstract.

  1. Please demonstrate locations of apoptotic microglial cells in P2X7 -/- or wild-type mice treated with sodium iodate injections regarding figure 3/4 using double staining with TUNEL and microglial markers.

Ans: Sorry due to the time restrain we cannot provide TUNEL data.

  1. As shown in figure 9, under the sodium iodate stimulation, please demonstrate how inflammatory cytokines in the supernatants derived/secreted from BV2 microglial cells involve cell death in cultured RPE cells.

Ans: It is well known that microglia play the most important role in innate immunity to sense and respond stress insults by upregulating proinflammatory gene expression and cytokine release.

  1. Finally, the authors should include recent review article looking at the role of P2X7 in retinal diseases (Biochem Pharmacol doi: 10.1016/j.bcp.2022.114942.Epub 2022 Feb 5) to ensure the novelty of this study.

Ans: Yes, we have included this reference as per your suggestion in ref 17.

Reviewer 3 Report

The article addresses the role of the P2X7 receptor in the retina and its relationship with retinal degeneration. The strategy and pipeline used by the authors are correct. The models (P2X7 KO mice and retinal cell lines) used are adequate, the experimental design is clever and research is well conducted. A great variety of techniques including molecular and cellular biology, physiology, imaging, etc., are used toward this end. The Discussion section is well elaborated and the conclusions drawn are backed up by the results. Yet, a series of minor points deserve the attention of the authors in order to improve the manuscript.

Major points

English needs a thorough revision, especially (though not exclusively) the use of articles (‘the’ and ‘a’). At some instances the article is missing while in others it must (or can) be omitted. For instance, the mice (throughout the ms.), the gene expression (line 233), the a significant attenuation (line 236), the immune system (line 356), a receptor antagonist (line 359), etc.

Minor points

The use of italics in the Abstract is not understood. Please avoid

It is somewhat contradictory that P2X7 is ubiquitously expressed (which means that is expressed in virtually all cell types) and in the same sentence particularize it for myeloid cells (lines 30-31). It is more appropriate to state that P27X is ubiquitously expressed in the mouse organism including myeloid cells.

Activation of P2X7 can trigger a process, not ions. The word ‘flow’ or ‘movement’ is missing in the sentence in line 35.

In the sentence ‘In the retina, P2X7 was found in both inner and outer cells including microglia’ (line 66), this referee believes that the authors refer to cells in the outer and inner retina. If so, please correct.

‘Gene expressions’ is incorrect. Please rewrite the sentence in lines 146-147 so as to read ‘The expression of genes encoding IL-1β… was quantitated by real-time PCR analysis’.

The antibodies used for immunoblotting and ELISA must be specified in Materials and Methods, including catalog numbers, dilutions, incubation times, etc.

Abcam is repeated in lines 166-167. Also, please specify that the NADPH concentration is measured spectrophotometrically.

NaIO3 injection is not ‘for 1 day’ (line 182), but at day 1, this referee believes. If agreed, please correct.

In line 209 it can be read ‘Some round, pigmented granules lysing along with RPE layer’. Do the authors mean ‘lying’?

In line 244 where it reads ‘we were interested to understand the caspase expression’ it should read ‘we were interested in understanding caspase expression’.

In line 268 where it reads ‘we wonder the action’ it should read ‘we wonder if the action’.

In line 370 ‘amplify’ must be substituted by ‘amplifies’, etc., etc. (please thoroughly revise the ms.)

In Fig. 6B, 3rd graph, an increase of the NADP/NADPH ratio can be seen upon addition of NaIO3, especially in the WT mice. Yet the authors claim that NaIO3 did not change the cellular levels ot total NADP, NADPH and ratio of NADP/NADPH. Have stats been appropriately made?

In Figure 9, RPE nuclei must be drawn much bigger and nearer to the cells’ basal surface. See e.g. Bonilha 2014 Exp. Cell Res. and/or the Webvision webpage.

Author Response

Reviewer 3

The article addresses the role of the P2X7 receptor in the retina and its relationship with retinal degeneration. The strategy and pipeline used by the authors are correct. The models (P2X7 KO mice and retinal cell lines) used are adequate, the experimental design is clever and research is well conducted. A great variety of techniques including molecular and cellular biology, physiology, imaging, etc., are used toward this end. The Discussion section is well elaborated and the conclusions drawn are backed up by the results. Yet, a series of minor points deserve the attention of the authors in order to improve the manuscript.

Major points

English needs a thorough revision, especially (though not exclusively) the use of articles (‘the’ and ‘a’). At some instances the article is missing while in others it must (or can) be omitted. For instance, the mice (throughout the ms.), the gene expression (line 233), the a significant attenuation (line 236), the immune system (line 356), a receptor antagonist (line 359), etc.

Ans: Thanks for the comments. We have revised our manuscript based on the comments.

Minor points

  1. The use of italics in the Abstract is not understood. Please avoid

Ans: Yes, we have revised it.

  1. It is somewhat contradictory that P2X7 is ubiquitously expressed (which means that is expressed in virtually all cell types) and in the same sentence particularize it for myeloid cells (lines 30-31). It is more appropriate to state that P27X is ubiquitously expressed in the mouse organism including myeloid cells.

Ans: We have corrected the sentences.

  1. Activation of P2X7 can trigger a process, not ions. The word ‘flow’ or ‘movement’ is missing in the sentence in line 35.

Ans: Thanks. We have corrected it, now it is in line 36.

  1. In the sentence ‘In the retina, P2X7 was found in both inner and outer cells including microglia’ (line 66), this referee believes that the authors refer to cells in the outer and inner retina. If so, please correct.

Ans: We have corrected it.

  1. ‘Gene expressions’ is incorrect. Please rewrite the sentence in lines 146-147 so as to read ‘The expression of genes encoding IL-1β… was quantitated by real-time PCR analysis’.

Ans: We have revised the line as per your suggestion.

  1. The antibodies used for immunoblotting and ELISA must be specified in Materials and Methods, including catalog numbers, dilutions, incubation times, etc.

Ans: We have revised as per your suggestion.

  1. Abcam is repeated in lines 166-167. Also, please specify that the NADPH concentration is measured spectrophotometrically.

Ans: We have revised as per your suggestion.

  1. NaIO3injection is not ‘for 1 day’ (line 182), but at day 1, this referee believes. If agreed, please correct.

Ans: We have revised as per your suggestion.

  1. In line 209 it can be read ‘Some round, pigmented granules lysing along with RPE layer’. Do the authors mean ‘lying’?

Ans: Yes, it was a typo. We revised it as “lying”.

  1. In line 244 where it reads ‘we were interested to understand the caspase expression’ it should read ‘we were interested in understanding caspase expression’.

Ans: We have corrected it shown in line 242.

  1. In line 268 where it reads ‘we wonder the action’ it should read ‘we wonder if the action’.

Ans: We have revised as per your suggestion.

  1. In line 370 ‘amplify’ must be substituted by ‘amplifies’, etc., etc. (please thoroughly revise the ms.)

Ans: We have revised as per your suggestion.

  1. In Fig. 6B, 3rd graph, an increase of the NADP/NADPH ratio can be seen upon addition of NaIO3, especially in the WT mice. Yet the authors claim that NaIOdid not change the cellular levels of total NADP, NADPH and ratio of NADP/NADPH. Have stats been appropriately made?

Ans: After checking, we confirmed our previous statistic results in Fig. 6B, meaning no statistic significance of increased NADP/NADPH ratio in NaIO3-treated mice in WT group.

  1. In Figure 9, RPE nuclei must be drawn much bigger and nearer to the cells’ basal surface. See e.g. Bonilha 2014 Exp. Cell Res. and/or the Web vision webpage.

Ans: We have revised the summary figure as suggested.

Round 2

Reviewer 2 Report

I strongly recommend that the reviewer's point "Please demonstrate locations of apoptotic microglial cells in P2X7 -/- or wild-type mice treated with sodium iodate injections regarding figure 3/4 using double staining with TUNEL and microglial markers." should be replied and the authors should examine additional experiments.

  1.  
  2.  
